



1    Evaluation of the Aqua MODIS Collection 6.1 multilayer cloud detection algorithm through

2                    comparisons with CloudSat CPR and CALIPSO CALIOP products

4                    Benjamin Marchant[1,2], Steven Platnick[1], Kerry Meyer[1], and Galina Wind[1,3]

5            1: NASA Goddard Space Flight Center; 2: USRA Universities Space Research Association; 3:

6                            SSAI: Science Systems and Applications, Inc.

7                                    benjamin.marchant@nasa.gov

**Abstract:**

11        Since multilayer cloud scenes are common in the atmosphere and can be an important

source of uncertainty in passive satellite sensor cloud retrievals, the MODIS MOD06/MYD06
standard cloud optical property products include a multilayer cloud detection algorithm to assist
with data quality assessment. This paper presents an evaluation of the Aqua MODIS MYD06
Collection 6.1 (C6.1) multilayer cloud detection algorithm through comparisons with active CPR
and CALIOP products that have the ability to provide cloud vertical distributions and directly
classify multilayer cloud scenes and layer properties. To compare active sensor products with an
imager such as MODIS, it is first necessary to define multilayer clouds in the context of their
radiative impact on cloud retrievals. Three main parameters have thus been considered in this
evaluation: (1) the maximum separation distance between two cloud layers, (2) the
thermodynamic phase of those layers, and (3) the upper layer cloud optical thickness. The impact
of including the Pavolonis-Heidinger multilayer cloud detection algorithm, introduced in Collection
6, to assist with multilayer cloud detection has also been assessed. For the year 2008, the MYD06
C6.1 multilayer cloud detection algorithm identifies roughly 20 percent of all cloudy pixels as





multilayer (decreasing to about 13 percent if the Pavolonis-Heidinger algorithm output is not
used). Evaluation against the merged CPR and CALIOP 2B-CLDCLASS-lidar product shows that
the MODIS multilayer detection results are quite sensitive to how multilayer clouds are defined in
the radar/lidar product, and that the algorithm performs better when the optical thickness of the
upper cloud layer is greater than about 1.2 with a minimum layer separation distance of 1km.
Finally, we find that filtering the MYD06 cloud optical properties retrievals using the multilayer
cloud flag improves aggregated statistics, particularly for ice cloud effective radius.

**I - Introduction**

Detection of multilayer clouds using passive sensors such as the Moderate-resolution

Imaging Spectroradiometer (MODIS) is a challenging but important remote sensing need. The
existence of multiple cloud layers may strongly impact retrievals of cloud optical, microphysical,
and cloud-top properties under single layer plane-parallel cloud assumptions. For example, the
MODIS Collection 6/6.1 (C6/C6.1) cloud optical property retrievals (MOD06/MYD06 for
Terra/Aqua, respectively), which assume a homogeneous plane-parallel cloud model as did
previous collections (Platnick et al. 2017), have been shown to have significant microphysical
cloud retrieval errors or outright failures for pixels that are identified as multilayer. As such, a
multilayer cloud detection algorithm (Wind et al. 2010) was first developed for Collection 5 as a
quality assurance metric to identify multilayer cloudy scenes.  The MYD06 multilayer cloud flag
has subsequently been used synergistically with optical centroid cloud pressure derived from
Ozone Monitoring Instrument (OMI) UV observations to further identify multilayer and vertically
extended clouds (Joiner et al. 2010). Beyond MODIS, other passive multilayer cloud detection
techniques use polarized reflectances, such as those from the Polarization and Directionality of
the Earth's Reflectance (POLDER) instrument (Desmons et al, 2017), in addition to spectral





signature differences between monolayer and multilayer cloud scenes determined from forward
radiative transfer models (Pavolonis and Heidinger, 2004; Heidinger and Pavolonis, 2005; Nasiri
and Baum, 2004; Jin and Rossow, 1997). Several studies have also been dedicated to the
inference of cloud optical properties for multilayer cloud scenes, e.g., Watts et al. (2011),
Sourdeval et al. (2014) and Chang and Li (2005). Those studies use a two-layer cloud overlapping
model approximation coupled with, e.g., optimal estimation, to derive the cloud optical properties
associated with the two cloud layers, and thus inherently require robust multilayer cloud detection.

Evaluating the performance of multilayer cloud detection algorithms requires appropriate

truth datasets and an understanding of the intent of the algorithm itself. For instance, the
MOD06/MYD06 multilayer cloud detection algorithm was initially evaluated using forward
radiative transfer simulations (Wind et al., 2010), though these cannot fully capture the complexity
of the real atmosphere. Active sensors, on the other hand, such as the CloudSat Cloud Profiling
Radar (CPR) and the Cloud-Aerosol Lidar with Orthogonal Polarization (CALIOP) onboard the
Cloud-Aerosol Lidar and Infrared Pathfinder Satellite Observation (CALIPSO) satellite, both in the
afternoon "A-train" constellation, provide key details on cloud vertical structure. Merged
CPR/CALIOP products that exploit the different yet complementary sensitivities of radar and lidar
observations have demonstrated utility for evaluating passive multilayer cloud detection
algorithms. In fact, the MOD06/MYD06 multilayer cloud flag previously has been evaluated by
Wang et al. (2016) using the 2B-CLDCLASS-LIDAR product for the years 2007-2010, and by
Desmons et al. (2017), who in parallel evaluated the PARASOL-POLDER multilayer cloud
detection algorithm using the 2B-GEOPROF-lidar and CALIOP 5km cloud layer products for the
years 2006-2010. These investigations, however, broadly defined multilayer clouds in the
radar/lidar datasets and thus implicitly did not consider the intent of the MOD06/MYD06 multilayer
cloud detection algorithm, which is to identify scenes where a second cloud layer adversely


impacts the optical property retrievals of the radiatively dominant cloud layer (the primary example
being a thin ice cloud overlying an optically thicker liquid water cloud), rather than as a strict
multilayer detection algorithm. For example, Desmons et al. defined a multilayer cloud when CPR
and CALIOP detected two spatially distinct cloud layers, regardless of the separation distance
between the cloud layers and cloud thermodynamic phase, while Wang et al. specified only that
detected cloud layers must be separated vertically by at least 480m to be considered multilayer.

In this paper, the main purpose is to present an evaluation of the Aqua MODIS (MYD06)
C6.1 multilayer cloud detection algorithm through comparisons with CPR and CALIOP merged
products. In addition, we will investigate how multilayer clouds affect MYD06 cloud
thermodynamic phase results which have strong consequences for microphysical retrievals. In
the first section we provide a short overview of the MOD06/MYD06 multilayer cloud detection
algorithm. The second section provides details about the datasets and the methodology used for
the evaluation. The third section presents evaluation results as a function of three main
parameters used to define a multilayer cloud scene in the CPR/CALIOP merged products: (1) the
separation distance $d$ between the two radiatively dominant cloud layers, (2) the thermodynamic
phase of those layers, and (3) the layer optical thicknesses, in particular of the upper cloud layer.
Finally, in the last section, we show the impact of multilayer clouds on cloud effective radius
retrievals.

**III – The MOD06/MYD06 multilayer cloud detection algorithm**

Originally introduced in Collection 5 (C5), the MOD06/MYD06 multilayer cloud detection
algorithm was developed as a quality assurance (QA) flag to identify scenes where the single-
layer cloud forward model assumption is likely violated. Its primary targets are those scenes



where an optically thinner cloud overlies an optically thicker liquid cloud, either where the phases
of the two layers differ (ice over liquid) or the vertical separation is sufficiently large such that
retrievals of the optical properties of the radiatively dominant underlying cloud are adversely
impacted. The algorithm operates on a pixel-level basis (1km resolution at nadir), with cumulative
results reported in the Cloud_Multi_Layer_Flag Science Data Set (SDS) in the MOD06/MYD06
Level-2 files and individual test results reported as bit values in the Quality_Assurance_1km SDS.
Full details on the C5 algorithm can be found in Wind et al. (2010); updates for C6/C6.1 are
summarized in Platnick et al. (2017) and the C6/C6.1 User's Guide (Platnick et al., 2018).

The algorithm is based primarily on four tests that are collectively used to classify a cloudy

pixel as monolayer or multilayer:
1. A cloud thermodynamic phase difference test, where divergent results between the IR

phase algorithm (Baum et al., 2012) and the shortwave/IR optical properties phase

algorithm (Marchant et al., 2016) yield a positive multilayer cloud result.

2. An above-cloud precipitable water (PW) difference test ($\Delta$PW), using the relative difference

between above-cloud PW derived from the $CO_2$-slicing cloud-top pressure result and that

derived from the $0.94\mu m$ channel with respect to the total PW (TPW) derived from ancillary

atmospheric profiles; a relative difference larger than 8% yields a positive multilayer cloud

result.

3. A second above-cloud PW difference test ($\Delta$PW$_{900mb}$), similar to the $\Delta$PW test above but

assuming the cloud is located at 900mb when deriving above-cloud PW from the $0.94\mu m$

channel; again, a relative difference of 8% yields a positive multilayer cloud result.

4. A test based on the algorithm of Pavolonis and Heidinger (2004) (hereafter referred to as

PH04 for brevity), introduced in C6, that uses reflectance at $0.65\mu m$ and 11 and $12\mu m$



brightness temperatures and brightness temperature differences, in addition to
reflectances at 1.6 and 1.38$\mu$m.

A test based on the divergence of cloud optical thickness retrievals from the standard VNSWIR
(Visible, near or shortwave infrared)-2.1$\mu$m channel pair and the 1.6-2.1$\mu$m channel pair was also
introduced in C6, but updates to the optical properties retrieval solution logic rendered this test
ineffective (see Platnick et al., 2018) and we do not consider it here. Note that the MOD06/MYD06
multilayer cloud algorithm is only applied to pixels having cloud optical thickness larger than 4.
Moreover, during algorithm development, the above tests, when positive, were assigned pre-
defined confidence values, the summation of which is reported in the Cloud_Multi_Layer_Flag
SDS and was intended to provide a pseudo-confidence level; a value of 0 indicates no cloud was
detected, 1 indicates a monolayer cloud, and values 2-10 indicate the cumulative weight of the
positive multilayer tests.

Figure 1 shows aggregated Aqua MODIS MYD06 Level 2 cloud products over the year 2008
(all data from C6.1 unless otherwise noted): (a) total cloud fraction from the MYD35 cloud mask
product after removing pixels identified as heavy aerosol or sun glint by the MYD06 clear sky
restoral (CSR) algorithm, (b) multilayer cloud fraction, (c) multilayer cloud fraction without the
PH04 test, and (d) C5.1 multilayer cloud fraction. The multilayer cloud fractions determined by
each individual C6/C6.1 multilayer cloud detection test are shown in the remaining panels: (e)
cloud phase difference test, (f) $\Delta$PW test, (g) $\Delta PW_{900mb}$ test, and (h) PH04 test. Note that the
multilayer fraction shown in Fig. 1c uses a similar definition for multilayer clouds, i.e., excluding
the PH04 test, as does the MOD08/MYD08 C6/C6.1 Level-3 (L3) aggregated products; this test
was excluded during C6 L3 development after preliminary analysis indicated that it was overly
aggressive in some circumstances. For the year 2008, we find that about 20% of cloudy pixels





are flagged as multilayer clouds, a number that decreases to 13% if the PH04 test is excluded
(similar to MOD06/MYD06 C5 results, Fig. 1d). Considering the multilayer cloud fraction in Fig.
1b where all tests contribute to the results, we find that about 21% of all positive multilayer cloud
results have a positive cloud phase difference test, 28% have a positive ΔPW test, 44% have a
positive $\Delta PW_{900mb}$ test, and 74% have a positive PH04 test.

**III - Data Sets and Methodology**

We evaluate the MODIS C6.1 multilayer cloud detection algorithm using co-located A-Train

CloudSat CPR and CALIPSO CALIOP data during the year 2008. Due to its location in the A-
Train, only Aqua MODIS MYD06 data is used; note that the multilayer algorithm applied to Terra
MODIS is identical to that applied to Aqua MODIS. Rather than consider CPR data separately,
we use the 2B-CLDCLASS-lidar CPR-CALIOP merged product in addition to the CALIOP Version
4 5km cloud layer products. The 2B-CLDCLASS-lidar product combines CPR and CALIOP
observations to provide cloud top and base heights jointly with cloud thermodynamic phase (ice,
liquid or mixed) for each cloud layer; Figure 2 shows an example 2B-CLDCLASS-lidar curtain for
a 2008-07-01 data segment starting at 01h 23min. This product provides up to 10 vertical cloud
layers at 1km horizontal resolution along-track. Since the upper cloud layer optical thickness is
critical in understanding the impact of multilayer cloud scenes on MYD06 cloud optical property
retrievals, cloud optical thickness from the CALIOP 5km layer product is merged with the
CLDCLASS-lidar product. This is accomplished by re-sampling the CALIOP product at 1km and
searching for matching cloud layers between the CALIOP 5km and 2B-CLDCLASS-lidar 1km
cloud layer products. Collocated files of MODIS and 2B-CLDCLASS-lidar have also been created
containing the pixel indices of 2B-CLDCLASS-lidar and the nearest MODIS pixel in terms of
spatial distance in the geographic coordinate system.




**IV - Evaluation of the MYD06 C6.1 multilayer cloud detection algorithm**


The global performance of the MYD06 multilayer cloud detection algorithm is shown in
Figure 3. Here, contingency tables comparing MYD06 multilayer classification results to those
from the 2B-CLDCLASS-lidar products are shown when the PH04 test is (a) included and (b)
excluded. Note that, for the 2B-CLDCLASS-lidar products, we use a naïve definition of multilayer
clouds here, namely all profiles where the merged product indicates more than one cloud layer
regardless of layer phase, optical thickness, or separation distance. Several conclusions can be
inferred from these tables. First, for the cloudy pixel population for which the MYD06 multilayer
detection algorithm is not applied (cloud optical thickness < 4, top rows), the 2B-CLDCLASS-lidar
product indicates a quite high percentage of multilayer clouds, 16.58% of the total cloudy
population. As we will show in the next section, this imposed multilayer detection limit in MYD06
can impact cloud effective radius retrieval statistics. For the cloudy pixel population for which the
MYD06 multilayer detection algorithm is applied (cloud optical thickness > 4, middle and bottom
rows), the MYD06 results including the PH04 test agree with the 2B-CLDCLASS-lidar monolayer
and multilayer classifications 33.73% of the time (21.29% for monolayer, 12.44% for multilayer),
and disagree 20.04% of the time (12.25% false multilayer detection rate, 7.79% false monolayer
detection rate). When the PH04 test is not included, the agreement and disagreement
percentages remain roughly the same, 34.95% and 18.83%, respectively, though the
apportionment between true/false mono/multilayer detection changes.

While it is evident in Figure 3 that MYD06 misses a relatively large percentage of multilayer
clouds that the radar/lidar merged product detects (7.79% or 11.40% when the PH04 test is
included or excluded, respectively), the active sensors are much more capable at detecting
multilayer cloud scenes than MODIS. More importantly, as we will see in the next section, in many
cases these missed multilayer scenes do not adversely impact the optical property retrieval
statistics and are thus beyond the intent of the algorithm. It is therefore important to evaluate the
algorithm's performance as a function of two parameters directly related to its intended targets,
namely the optical thickness of the upper layer cloud and the vertical separation distance of the
cloud layers.

To better understand the multilayer cloud scenes, we focus on multilayer cloud scenes with

only two cloud layers (which represent about 77% of the multilayer cloud population in our co-
located dataset). Figure 4 shows the probability that MYD06 correctly identifies a multilayer cloud,
using the 2B-CLDCLASS-lidar data as truth, given the separation distance $d$ (the distance
between the cloud base of the upper cloud and the cloud top of the bottom cloud) and the upper
layer cloud optical thickness $\tau$ defined by the CALIOP 5km cloud layer products. Results are
shown when including (a) and excluding (b) the PH04 test. Note that all 2B-CLDCLASS-lidar
multilayer cloud scenes are included in the baseline here regardless of layer thermodynamic
phase. One can see, from Figure 4a, that the PH04 test is very sensitive to multilayer clouds,
even if $d$ and $\tau$ are quite small, but at the expense of a larger false positive rate (see Figure 3a).
On the other hand, if the PH04 test is not used (Figure 4b), one can see that the probability of
correctly detecting a multilayer cloud scene increases with both d and $\tau$. Regardless of the
inclusion of the PH04 test, however, the results shown here indicate that it is probable that MYD06
will detect a multilayer cloud if the separation distance $d$ is greater than 1km and the upper layer
cloud optical thickness is greater than about 1.2.



In addition to cloud layer detection, the 2B-CLDCLASS-lidar products also provide a cloud
thermodynamic phase classification, i.e., liquid, ice or mixed phase, for each detected cloud layer
that can be used to evaluate the performance of the MYD06 cloud optical properties phase
algorithm in multilayer scenes. Note that the C6/C6.1 MOD06/MYD06 phase algorithm was tuned
and validated against the CALIOP 1 and 5 km cloud layer products using two months of collocated
data, though only for scenes where CALIOP observed only a single phase in the profile (Marchant
et al., 2016). Figure 5a shows a similar single-phase validation using the 2B-CLDCLASS-lidar
products for monolayer clouds only with a single cloud phase in 2008. While agreement for liquid
and ice phase results is 65.22%, 26.62% of 2B-CLDCLASS-lidar monolayer clouds are identified
as mixed phase, of which MYD06 identifies 9.85% and 16.77% as ice and liquid phase,
respectively.

Extending this monolayer analysis to multilayer cloud scenes, two types of multilayer cases
can be distinguished in the 2B-CLDCLASS-lidar product, namely profiles where the multiple cloud
layers share the same thermodynamic phase and those where they do not. Figure 5b shows the
comparison between the MYD06 cloud optical properties phase and the 2B-CLDCLASS-lidar
product for two cloud layers sharing the same cloud phase (roughly 10% of the co-located
dataset). When 2B-CLDCLASS-lidar identifies two ice layers or two liquid layers in the profile, the
MYD06 phase agrees 82.57% of the time. However, in 12.06% of the multilayer cases, MYD06
misidentifies an ice cloud overlapping another ice cloud as liquid cloud phase.

Figure 6 shows phase comparison results for the cases where 2B-CLDCLASS-lidar
identifies two cloud phases in the vertical profile (roughly 20% of the co-located dataset). The
most frequent cloud scene is an ice cloud overlapping a liquid cloud (59.54% of these cases, first
column), for which MYD06 identifies fractions of 27.27% ice and 32.27% liquid clouds. For ice



clouds overlapping mixed phase clouds, the second most frequent scene (30.74% of these cases,
second column), MYD06 is more likely to identify ice phase (16.45%) rather than liquid phase

(14.29%).


The ambiguity of the results in Figure 6 underscores the difficulty of determining a single
phase in a multilayer scene using MODIS when there is no unique answer on the true column
phase. Moreover, because the MYD06 cloud optical properties phase is a radiatively derived
designation, it must depend on, for example, the upper layer cloud optical thickness and the
sun/satellite viewing geometry. Focusing only on the case of ice clouds overlapping liquid clouds,
Figure 7 shows the probability that MYD06 (a) correctly identifies a multilayer cloud (PH04 test
excluded), and the probabilities of (b) undetermined, (c) ice, and (d) liquid phase results, each as
a function of layer separation distance $d$ and upper layer cloud optical thickness $\tau$. The probability
that MYD06 correctly identifies an ice cloud overlapping a liquid cloud as multilayer (Fig. 7a) is
similar in pattern to the probabilities for all multilayer scenes regardless of the cloud layer phase
in Figure 4b, though the magnitude of the probabilities here is larger. The MYD06 phase result
probabilities (Fig. 7b-d) are largely what one would expect, in particular that the probability of an
ice cloud result increases as the upper ice cloud optical thickness increases, while the probability
of a liquid cloud result shows the opposite pattern; the probability of an undetermined phase result
is largest when the two cloud layers are vertically close and the upper layer cloud optical thickness
is greater than 0.7.

**V - Assessing the MYD06 multilayer cloud flag as an optical property retrieval quality**
**indicator**





Given the intent of the MOD06/MYD06 multilayer cloud detection algorithm, namely to
identify scenes that do not conform to the single-layer cloud forward model assumption, we
assess the utility of the multilayer algorithm's results as a QA tool for the cloud optical property
retrievals. In particular, we focus on cloud effective radius retrievals, where multilayer scenes are
expected to have retrieval artifacts or uninterpretable results due to the mixing of particle
scattering properties from multiple cloud layers having different phases and/or microphysics. To
facilitate the analysis, we again use the collocated MYD06 and 2B-CLDCLASS-lidar 2008 dataset,
and consider two cloudy pixel populations: (1) a reference population containing only monolayer
clouds as determined by the 2B-CLDCLASS-lidar product for which the cloud thermodynamic
phase is in agreement with that of MYD06; (2) a population of multilayer clouds, defined as those
for which  the 2B-CLDCLASS-lidar product identifies more than one cloud layer regardless of the
cloud layer separation distance, the upper layer cloud optical thickness, or the cloud
thermodynamic phase.

Figure 8 presents the results for liquid (left column) and ice (right column) clouds for the
three primary cloud effective radius retrievals reported in the MYD06 cloud optical products,
namely those associated with three particle absorptive bands at 2.1, 1.6 and 3.7$\mu$m. One can see
the differences between the monolayer cloud (blue) and multilayer cloud (red) populations, and
that the ice cloud effective radius populations exhibit the largest differences. In particular, the ice
cloud effective radius distributions for the multilayer cloud population have a secondary mode at
effective radius around 10-15$\mu$m. This secondary mode can be explained by a large fraction of
cases in the co-located dataset having ice overlapping liquid clouds (see Figure 6, left column).
Since liquid droplets are less absorptive than ice crystals in these spectral channels for a given
size, identifying these scenes as ice phase can yield smaller ice cloud effective radius retrievals.
Indeed, if we remove from the multilayer population those cloudy pixels classified by MYD06 as



multilayer, as shown in Figure 9 for cases where MYD06 cloud optical thickness exceeds 4, one
can see that the secondary peaks in the ice effective radius distributions for multilayer clouds
(red) have disappeared. Therefore, though the MYD06 multilayer cloud detection is not able to
detect all multilayer clouds, it can be used to filter cloud effective radius retrievals that are
radiatively impacted by multilayer cloud scenes. Even if the PH04 algorithm is ignored in the
MYD06 multilayer cloud detection algorithm (Figure 10), the multilayer detection results remain
useful for removing most of the differences between the two populations, though some portion of
the small ice cloud effective radii remain.

If the MODIS cloud optical thickness is lower than 4, the multilayer cloud detection algorithm

is not applied since forward modeling indicated that there is not enough information to discriminate
monolayer and multilayer clouds (Wind et al. 2010). Figure 11 shows, however, that some
noticeable differences remain in the MODIS cloud effective radius distributions for monolayer and
multilayer clouds as determined by the 2B-CLDCLASS-lidar products. It is then not possible to
directly screen out the cloud effective radius strongly biased by the presence of multilayer cloud
scenes as we showed previously.

**VI – Conclusions**

In an evaluation of the MODIS multilayer cloud detection algorithm by comparing Aqua

MODIS MYD06 C6.1 with a merged CPR and CALIOP products, . As expected, the results are
quite sensitive to the definition of a multilayer cloud scene for active sensor products. Therefore,
three main parameters have been used to defined a multilayer cloud scene: (1) the maximum
separation distance $d$ between the two cloud layers, (2) the thermodynamic phase of those layers,
and (3) the upper layer optical thicknesses. Overall, the global MODIS multilayer cloud detection





algorithm skill performs well when the optical thickness of the upper layer is greater than about 1-
2 and the separation distance $d$ is greater than 1km. In parallel, the impact of using a 1.38 $\mu m$
channel in a multilayer algorithm (PH04, Pavolonis and Heidinger, 2004) was studied; PH04 was
added as a separate test to the MODIS multilayer algorithm beginning with Collection 6. It was
found that this algorithm flags too many cloudy scenes as multilayer, leading to an increase in
false positive occurrences, i.e. cloudy pixels wrongly flagged as multilayer.

This study also allowed for an expanded evaluation of the MODIS cloud

thermodynamic phase (Marchant et al. 2016), that was based on single layer CALIOP
observations, to the more general case of multilayer cloud scenes. For monolayer clouds, the
current analysis based on CPR and CALIOP gives results similar to Marchant et al. (which used
a different time period) in terms of showing a phase agreement fraction of about 91%. For two
spatially separated cloud layers detected by the CPR and CALIOP sensors, scenes with the same
cloud phase in the two layers were analyzed separately from scenes having different layer
phases. When the cloud phase is liquid in both cloud layers, there is good agreement between
the MODIS and active sensor cloud phases. When an ice cloud layer overlies another ice layer,
the MODIS phase is often retrieved as liquid; further investigation is needed for these cases.
When the cloud phase is different in the two cloud layers, the preferred phase for MODIS should
be based on the radiative contribution from each layer to the observed signal. For instance, the
most frequent cases, according to 2B-CLDCLASS-lidar products, are ice overlying liquid clouds
for which the fraction of ice or liquid cloud retrieved by MODIS are about the same but this includes
radiatively thin upper cloud layers. MYD06 is more and more likely to identify ice phase rather
than liquid phase with the increase of the ice cloud optical thickness.

Even though the MODIS C6 multilayer cloud detection algorithm is not able to detect all

multilayer cloud scenes compared to the merged CPR and CALIOP product (MYD06 results





including the PH04 test agree with the 2B-CLDCLASS-lidar monolayer and multilayer
classifications 33.73% of the time, disagree 20.04% of the time), the algorithm is reasonably
skilled in its intended use, i.e., discriminating those pixels for which the cloud effective radius may
be biased by layers having different microphysics (phase and/or effective particle size). MODIS
ice phase categorized clouds have effective radius retrievals that are most impacted by multilayer
cloud scenes, with a small radius bias. If the PH04 detection algorithm output is not used, the
fraction of multilayer clouds flagged by MODIS is smaller but the MODIS multilayer cloud
algorithm then has less skill to screen out cloud effective radius impacted by the presence of
multilayer clouds. Finally, if was found that when the column cloud optical thickness is less than
4, cutoff used by the MODIS algorithm, cloud effective radius retrievals can still be impacted by
multilayer clouds identified with the active sensor products. Further work on extending the MODIS
multilayer cloud detection algorithm to smaller column cloud optical thicknesses is warranted.

**V - References**

- Chang, F.-L., Li, Z.: A New Method for Detection of Cirrus Overlapping Water Clouds and

Determination of Their Optical Properties. Journal of the Atmospheric Sciences, 62(11), 3993–

4009. https://doi.org/10.1175/jas3578.1, 2005.

- Cho, H.-M., Zhang, Z., Meyer, K., Lebsock, M., Platnick, S., Ackerman, A. S., Girolamo, L.D. ,

Labonnote, L.C., Cornet, C., Riedi, J., E. Holz, R.E.: Frequency and causes of failed MODIS

cloud property retrievals for liquid phase clouds over global oceans. Journal of Geophysical

Research: Atmospheres, 120(9), 4132–4154. https://doi.org/10.1002/2015jd023161, 2015.

- Desmons, M., Ferlay, N., Parol, F., Riédi, J., Thieuleux, F.: A Global Multilayer Cloud

Identification with POLDER/PARASOL. Journal of Applied Meteorology and Climatology, 56(4),

1121–1139. https://doi.org/10.1175/jamc-d-16-0159.1, 2017.



- Heidinger, A. K., Pavolonis, M. J.: Global Daytime Distribution of Overlapping Cirrus Cloud
from NOAA's Advanced Very High-Resolution Radiometer. Journal of Climate, 18(22), 4772–
4784. https://doi.org/10.1175/jcli3535.1, 2005.
- Jin, Y., Rossow, W. B.: Detection of cirrus overlapping low-level clouds. Journal of Geophysical
Research: Atmospheres, 102(D2), 1727–1737. https://doi.org/10.1029/96jd02996, 1997.
- Joiner, J., Vasilkov, A. P., Bhartia, P. K., Wind, G., Platnick, S., Menzel, W. P.: Detection of
multi-layer and vertically-extended clouds using A-train sensors. Atmospheric Measurement
Techniques, 3(1), 233–247. https://doi.org/10.5194/amt-3-233-2010, 2010.
- Li, J., Huang, J., Stamnes, K., Wang, T., Lv, Q., Jin, H.: A global survey of cloud overlap based
on CALIPSO and CloudSat measurements. Atmospheric Chemistry and Physics, 15(1), 519–
536. https://doi.org/10.5194/acp-15-519-2015, 2015.
- Marchant, B., Platnick, S., Meyer, K., Arnold, G. T., Riedi, J.: MODIS Collection 6 shortwave-
derived cloud phase classification algorithm and comparisons with CALIOP. Atmospheric
Measurement Techniques, 9(4), 1587–1599. https://doi.org/10.5194/amt-9-1587-2016, 2016.
- Nasiri, S. L., Baum, B. A.: Daytime Multilayered Cloud Detection Using Multispectral Imager
Data. Journal of Atmospheric and Oceanic Technology, 21(8), 1145–1155.
https://doi.org/10.1175/1520-0426(2004)021<1145:dmcdum>2.0.co;2, 2004.
- Pavolonis, M. J., Heidinger, A. K.: Daytime Cloud Overlap Detection from AVHRR and VIIRS.
J. Appl. Meteorology, 43, 762-778, doi:10.1175/2099.1, 2004.
- Platnick, S., Meyer, K. G., King, M. D., Wind G., Amarasinghe N., Marchant B., Arnold G.T.,
Zhang Z., Hubanks P. A., Holz R.E., Yang P., Ridgway W. L., Riedi, J.: Te MODIS cloud optical
and microphysical products: Collection 6 updates and examples from Terra and Aqua. IEEE
Trans. Geosci. Remote Sens., 55, 502-525, 2017.



- Sassen, K., Wang, Z., Liu, D.: Global distribution of cirrus clouds from CloudSat/Cloud-Aerosol
Lidar and Infrared Pathfinder Satellite Observations (CALIPSO) measurements. Journal of
Geophysical Research, 113. https://doi.org/10.1029/2008jd009972, 2008.
- Sourdeval, O., C.-Labonnote, L., Baran, A. J., Brogniez, G.: A methodology for simultaneous
retrieval of ice and liquid water cloud properties. Part I: Information content and case study.
Quarterly Journal of the Royal Meteorological Society, 141(688), 870–882.
https://doi.org/10.1002/qj.2405, 2014.
- Sourdeval, O., C.-Labonnote, L., Baran, A. J., Mülmenstädt, J., Brogniez, G.: A methodology
for simultaneous retrieval of ice and liquid water cloud properties. Part 2: Near-global retrievals
and evaluation against A-Train products. Quarterly Journal of the Royal Meteorological Society,
142(701), 3063–3081. https://doi.org/10.1002/qj.2889, 2016.
- Vitter, J. S.: Random sampling with a reservoir. ACM Transactions on Mathematical Software,
11(1), 37–57. https://doi.org/10.1145/3147.3165, 1985.
- Wang, T., Fetzer, E. J., Wong, S., Kahn, B. H., Yue, Q.: Validation of MODIS cloud mask and
multilayer flag using CloudSat-CALIPSO cloud profiles and a cross-reference of their cloud
classifications. Journal of Geophysical Research: Atmospheres, 121(19), 11,620-11,635.
https://doi.org/10.1002/2016jd025239, 2016.
- Wang, Z., Vane D., Stephens G., Reinke D. Level 2 Combined Radar and Lidar Cloud Scenario
Classification Product Process Description and Interface Control Document
http://www.cloudsat.cira.colostate.edu/sites/default/files/products/files/2B-CLDCLASS-
LIDAR_PDICD.P_R04.20120522.pdf
- Watts, P. D., Bennartz, R., Fell, F.: Retrieval of two-layer cloud properties from multispectral
observations using optimal estimation. Journal of Geophysical Research, 116(D16).
https://doi.org/10.1029/2011jd015883, 2011.





418 - Wind, G., Platnick, S., King M.D., Hubanks, P.A., Pavolonis, M.J., Heidinger, A.K., Yang P.,

419  Baum, B.A.: "Multilayer Cloud Detection with the MODIS Near-Infrared Water Vapor Absorption

420  Band." Journal of Applied Meteorology and Climatology 49 11 (November): 2315–2333.

421  doi:10.1175/2010jamc2364.1. http://dx.doi.org/10.1175/2010JAMC2364.1, 2010.




Figure 1: A collection of aggregated (all pixel) Aqua MODIS Level 2 cloud products over the year 2008: (a) cloud fraction, (b) C6.1 multilayer cloud fraction, (c) C6.1 multilayer cloud fraction excluding the Pavolonis and Heidinger (2004) (PH04) test, and (d) C5.1 multilayer cloud fraction; fractions determined from each individual C6.1 multilayer cloud detection test: (e) cloud phase difference test, (f) ΔPW test (g) $\Delta PW_{900mb}$ test, and (h) PH04 test.














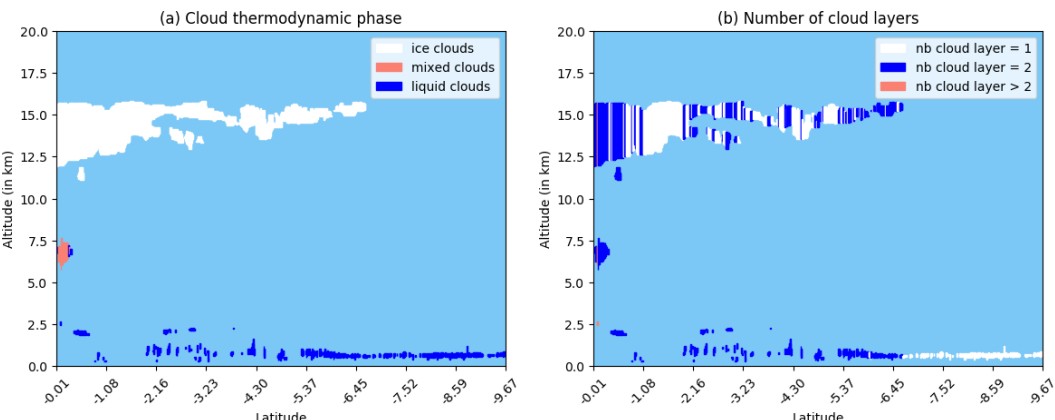

*Figure 2: An example 2B-CLDCLASS-lidar curtain (2008183012329_11573_CS_2B-CLDCLASS-LIDAR_GRANULE_P_R04_E02.hdf): (a) cloud thermodynamic phase for each detected cloud layer (ice, liquid or mixed); (b) the number of cloud layers found after merging cloud layers with a vertical separation distance less than 3km.*


















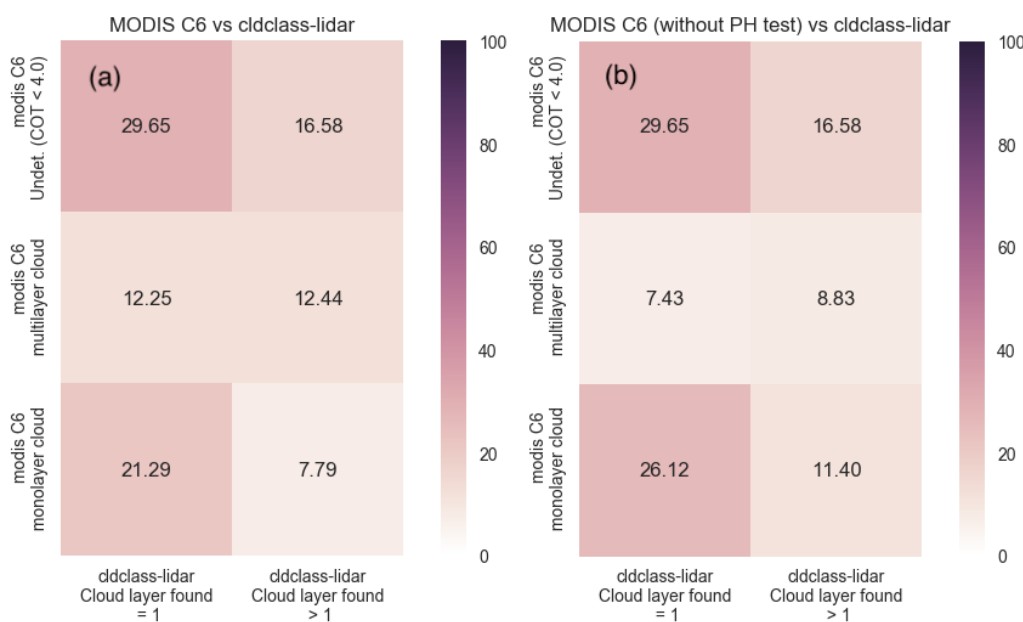

Figure 3: Contingency tables of the MYD06 C6.1 multilayer cloud detection algorithm compared against multilayer clouds defined by the 2B-CLDCLASS-lidar products: MYD06 (a) with and (b) without the Pavolonis-Heidinger (PH04) test. The 2B-CLDCLASS-lidar multilayer clouds are defined regardless of the separation distance between the cloud layers, the cloud thermodynamic phase or the cloud optical thickness.














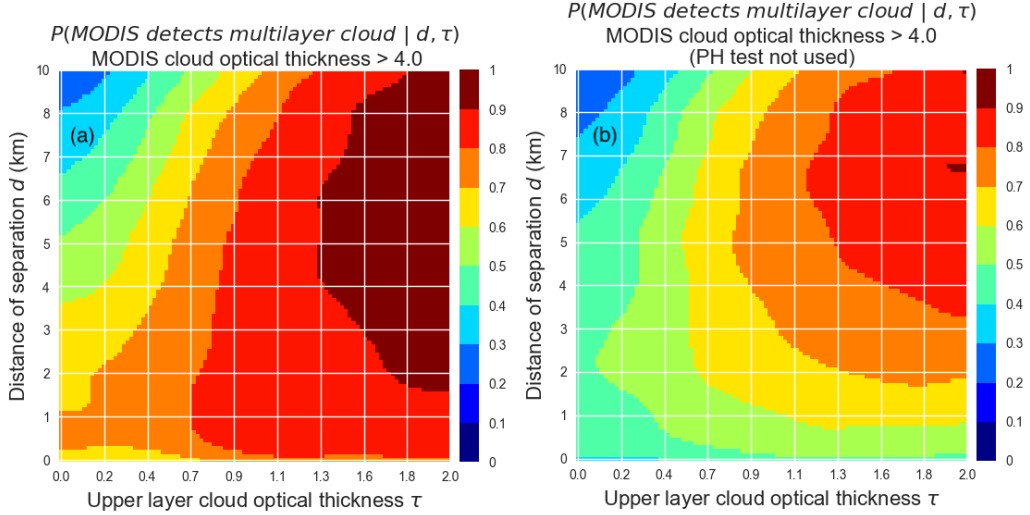

Figure 4: Probabilities that MYD06 detects a multilayer cloud, (a) with and (b) without the Pavolonis-Heidinger (PH04) test, given the separation distance between two cloud layers and the cloud optical thickness of the upper layer derived from 2B-CLDCLASS-lidar and CALIOP 5km cloud products, respectively.


















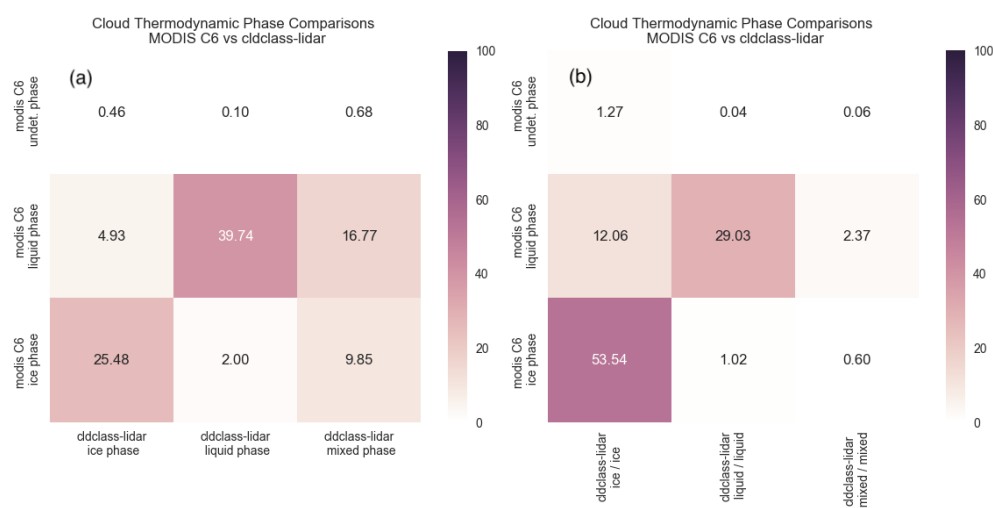

*Figure 5: MYD06 C6.1 cloud thermodynamic phase compared to 2B-CLDCLASS-lidar cloud phase: (a) monolayer clouds (about 63% of the dataset), and (b) multilayer clouds having the same phase (about 10% of the co-located dataset). Here, mono/multilayer clouds are defined by 2B-CLDCLASS-lidar.*


















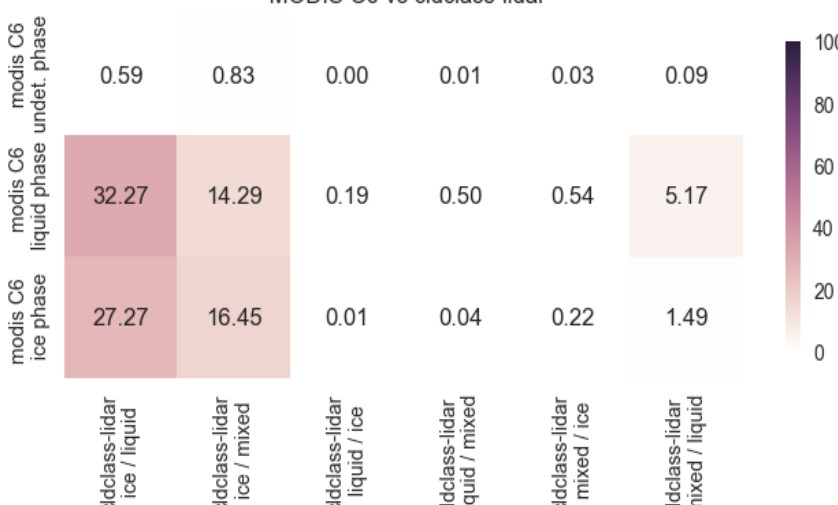

*Figure 6: MYD06 C6.1 cloud optical properties thermodynamic phase compared to 2B-CLDCLASS-lidar cloud phase for multilayer clouds having a different cloud phase in the vertical profile. "Ice/liquid" refers to an upper ice layer overlying a liquid cloud layer, and similarly for other notions (about 20% of the co-located dataset).*













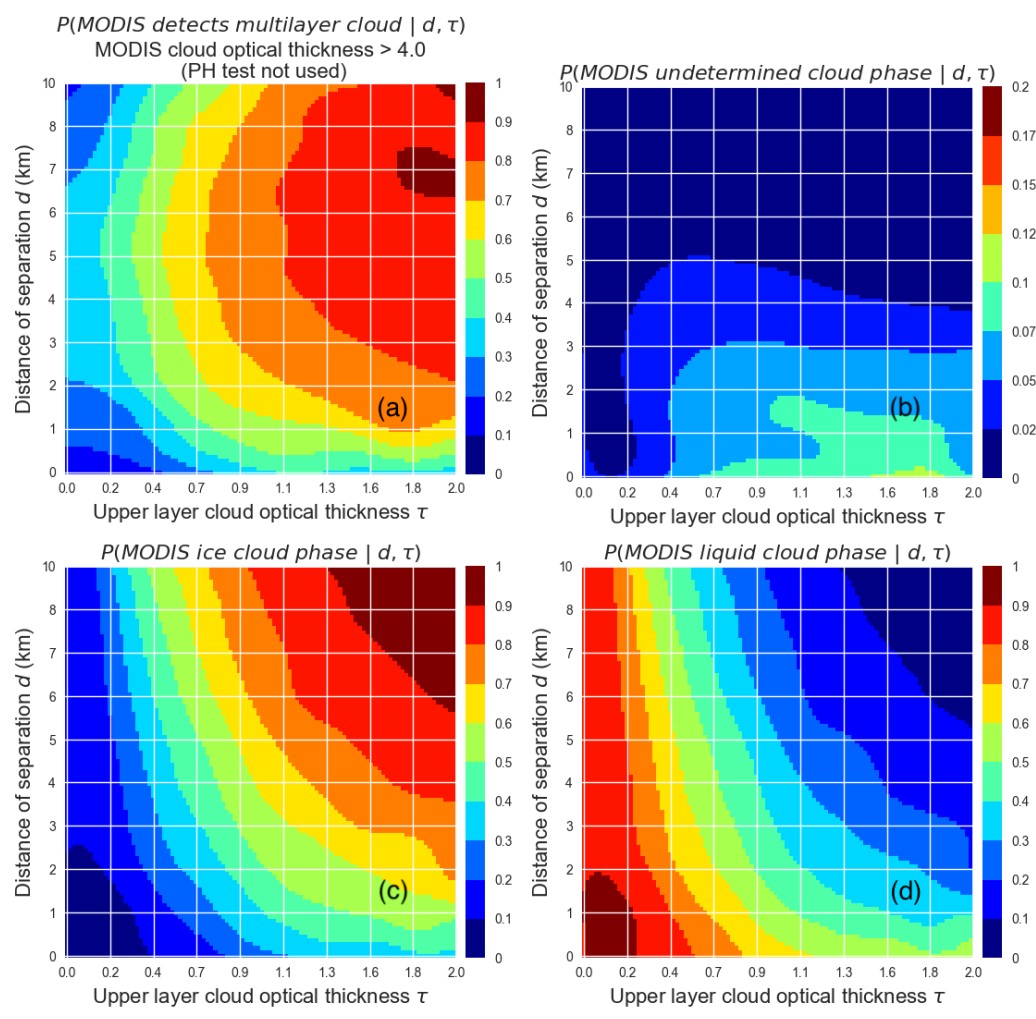

*Figure 7: (a) Probability that the MYD06 multilayer cloud detection algorithm detects an ice cloud overlapping a liquid cloud (with the PH test turned off) given the separation distance "d" between the two cloud layers and the upper layer cloud optical thickness "τ" defined by the 2B-CLDCLASS-lidar products; probabilities that the MYD06 cloud optical properties phase algorithm provides an undetermined (b), ice (c) or liquid (d) cloud phase given "d" and "τ".*


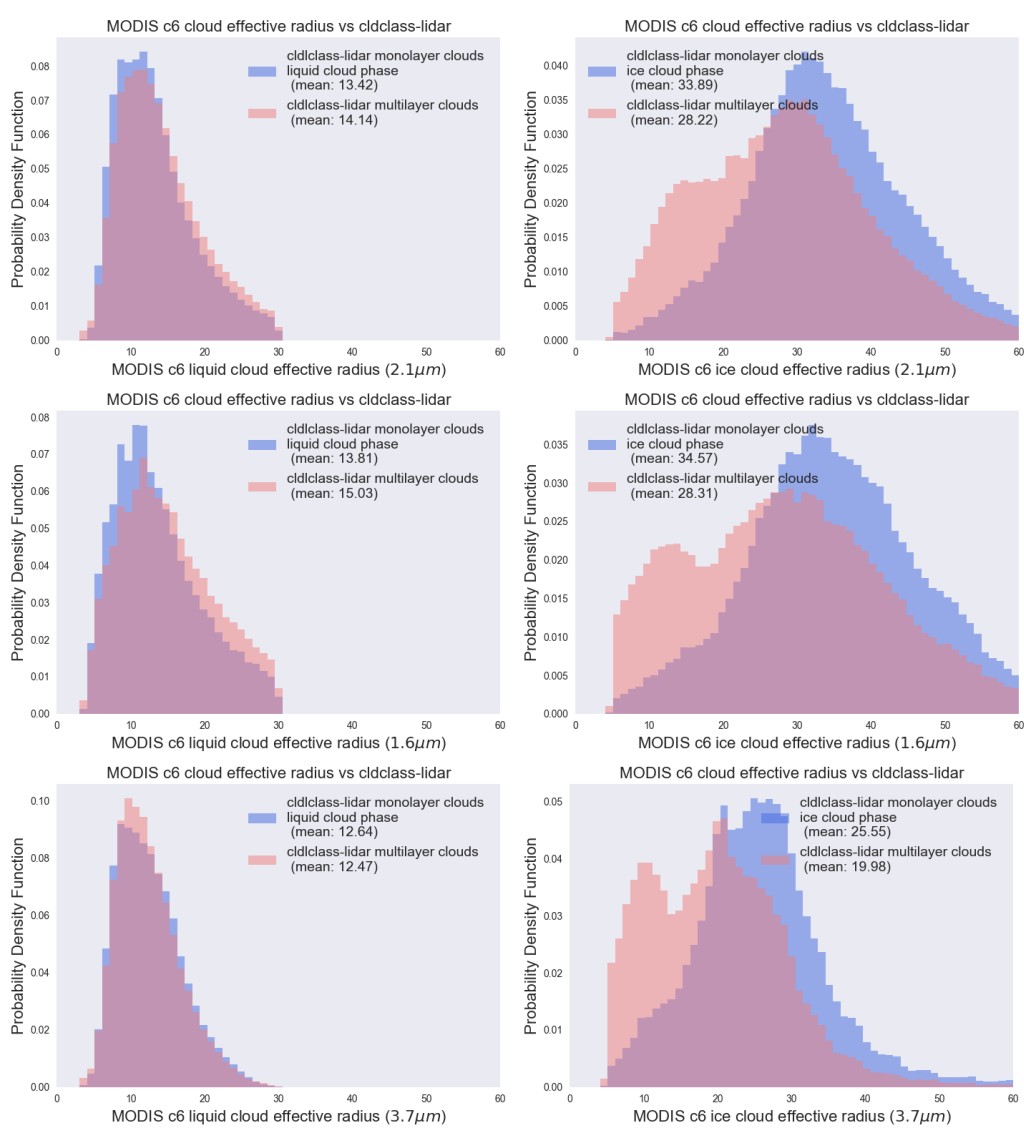

*Figure 8: MYD06 1.6, 2.1, 3.7 μm liquid (left column) and ice (right colum) cloud effective radius retrieval distributions for monolayer (light blue) and multilayer (light red) cloud populations as determined by the 2B-CLDCLASS-lidar products regardless of the cloud layer separation distance or the upper layer cloud optical thickness.*








Figure 9: Same as Figure 8, but for the population having MYD06 cloud optical thickness larger than 4 and after removing from the multilayer cloud population (in red) the cloudy pixels classified by the MYD06 multilayer cloud detection algorithm as multilayer clouds.













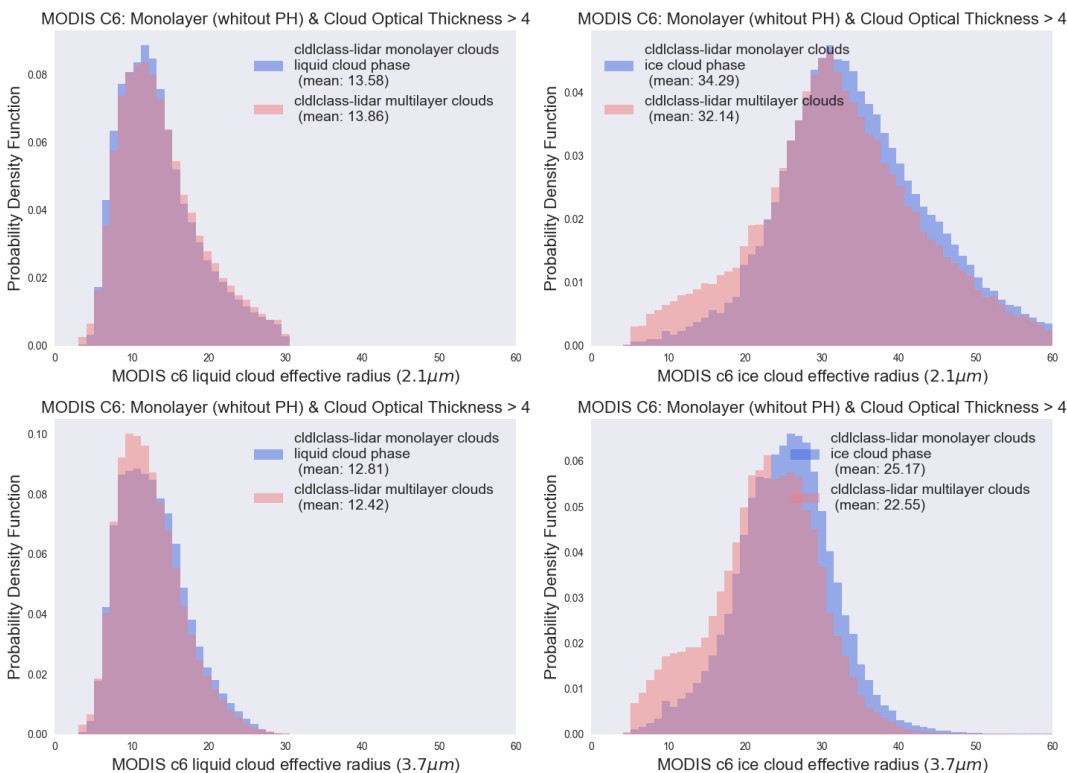

Figure 10: Same as Figure 9, but excluding the Pavolonis and Heidinger detection algorithm in the MYD06 multilayer cloud detection algorithm.














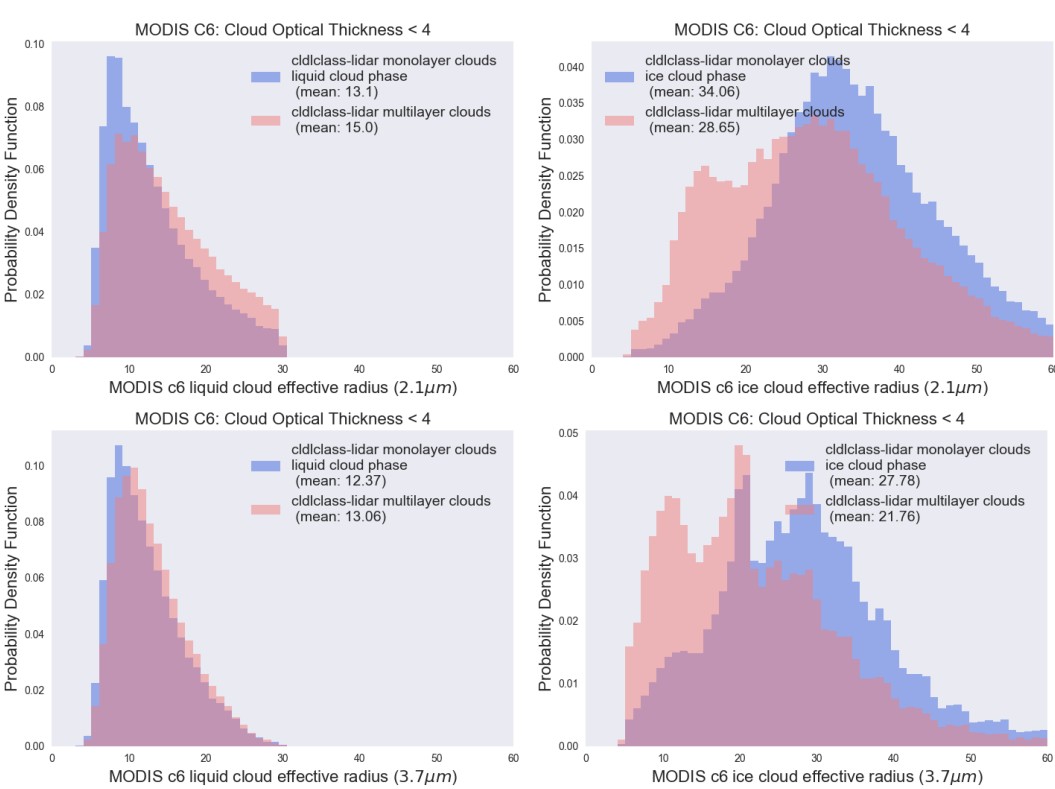

*Figure 11: Differences in MYD06 cloud effective radius distributions for monolayer (in blue) and multilayer (in red) clouds for the population having MYD06 cloud optical thickness lower than 4.*


