# Peer review of "Evaluation of the Agua MODIS Collection 6.1 multilayer cloud detection algorithm through 1 2 comparisons with CloudSat CPR and CALIPSO CALIOP products 3 4 Benjamin Marchant1,2, Steven Platnick1, Kerry Meyer1, and Galina Wind1,3 5 1: NASA Goddard Space Flig"

_Atmospheric Measurement Techniques, 2019_

## Referee Comment (RC1) · Anonymous Referee #2 · 13 Feb 2020

The manuscript addresses the fidelity of the Aqua MODIS Collection 6.1 multilayer cloud detection product against coincident observations from CloudSat and CALIPSO matched to the MODIS pixels. The MODIS multilayer product is a combination of four different tests, one of which depends on the discrepancy of thermodynamic phase derived from the mid-infrared and the shortwave infrared, two of which depend on ancillary water vapor products or derived column water vapor products, and the last based on the algorithm of Pavolonis and Heidinger. The CloudSat/CALIPSO combined 2B-cldclass-lidar product is used for ground truth of multilayer clouds and the phase

determination of each cloud layer detected. The 5-km CALIPSO cloud layer product is used to obtain cloud optical thickness for the different features detected and is over-sampled to 1-km for comparison purposes. The comparisons are stratified by whether single or multilayer clouds are detected, whether the optical thickness according to MODIS is greater than or less than four, or whether the multiple layer clouds are of the same or different phases. Histograms of cloud effective radius are shown for single and multiple layer clouds, and also for the different tests used for detecting multiple layer clouds. The histograms show that the purpose of the MODIS multilayer flag is sufficient to detect most outliers in the retrieved cloud effective radius product.

This is a useful study that dives into the details of the performance of the MODIS multilayer algorithm and the results are generally presented well. One area of the manuscript that left me a bit surprised is how often the multilayer flag of MODIS detects multiple cloud layers while the radar+lidar combination does not (see Figure 3, left panel, left column, middle row: 12.25%). My suspicion is that the disagreement between MODIS and the active sensors isn't nearly as poor as one might take away from these results because there could be some underlying sensitivity to the multi-layer tests to cloud vertical structure within single contiguous layers. In other words, the cloud water content could vary with altitude within a single layer but could trigger the multilayer flag in MODIS. One additional comparison that might be worth adding is some stratification based on how vertically homogeneous the cloud water content profiles are within single layers, and whether these are related to MODIS multilayer detections when C/C detects a single layer. CloudSat provides two flavors of cloud water content profiles (2B-CWC-RO and 2B-CWV-RVOD), and there is one flavor of a combined CloudSat/CALIPSO ice water content profile (2C-ICE). Perhaps these products could be stitched together on a profile-by-profile basis to make a single vertical profile of combined liquid and ice CWC? Perhaps the clouds that have a strong dependence of CWC on altitude within the layer have a multilayer flag that behaves differently than those that do not? I haven't thought through every technical detail to do this, but if feasible, I believe it is worth doing.

Otherwise, I only have minor suggestions for the revised version, and they are listed below.

The 'intent' of the algorithm is touched on at line 59 and lines 97-99. The latter occurrence seems out of place and would fit better near line 59. In fact, the 'intent' should be articulated for other multilayer algorithms besides MODIS.

Lines 77 and 79, references should have years added

Line 95, section should be two, not three

The discussion of figure 1 starting at line 138 is a little bit disjointed. I wasn't sure if panel (b) should be the sum of panels (e) to (h), or whether multiple positive tests can occur in a single pixel. I'm pretty sure it's the latter but it needs to be laid out clearer than is.

To be clear, the Pavolonis and Heidinger algorithm is available within the L2 products but not in the L3 products? If that is correct, why is that the case?

line 212, (a) and (b) should appear before including and excluding, respectively

line 252, answer about

figures 4 and 7 appear to have problematic axes. The number spacing in both axes is not uniform. Perhaps there is a rounding issue at play or the axes need to have additional bins or tick marks.

Figures 8 to 11, would be helpful to make clearer in each column at the top that this is "liquid" and "ice", or perhaps "liquid 2.1 um", "liquid 1.6 um", etc. The subpanel titles are pretty useless and could be included in the figure caption.

Furthermore, it would be easier to read the paper if cloud optical thickness reduced to the tau symbol or COT, and likewise with cloud effective radius could be $r_e$ or CER

---

## Referee Comment (RC2) · Anonymous Referee #3 · 22 Feb 2020

**Review of the paper: Evaluation of the Aqua MODIS Collection 6.1 multilayer cloud detection algorithm through comparisons with CloudSat CPR and CALIOP products**

**1. General comments**

This article evaluates the MODIS Collection C6.1 multilayer cloud detection algorithm through comparisons with CPR and CALIOP products.
This study is very interesting as it presents a thorough study of the MODIS ML algorithm performances. Indeed, it describes in details the importance of the cloud optical thickness, the vertical separation distance of the layers and the cloud thermodynamic phase for the identification of ML clouds as well as the consequences of those clouds on the MODIS retrievals.

Overall, I only have minor suggestions for the revision process.

**2. Specific comments**

section III: It's not clear to me if there are some changes in the MODIS ML algorithm between C6 and C6.1
although it might be worth to explain somewhere briefly the differences between the 2 collections

l240: About the ML clouds ice/ice identified as liquid by MODIS, do you have an idea why?

l322: In the end, would you recommend to keep this PH04 test for the MODIS ML algorithm?

Description of Fig2: is the product shown on Fig2b an official product? You do not mention or describe it in the paper. Is the 3km distance a common threshold to identify different layers?

**3. Technical corrections**

l37: ...layers **may** strongly...  replace by can, we are sure the presence of ML clouds can impact the retrievals

l49: I think the POLDER ML detection technique uses polarized reflectances but is not based on them.

l84-85: the sentence is not nice.

l107: and **in** the C6/C6.1

**Globally:** when you write 0.94 μm, like l20, there should be a space between the number and the unit, in Latex there is something similar to half a space (\, for me)

l123 to 125: not clear, do you mean : reflectances at 0.65 μm, 1.6 μm, 1.38 μm as well as brightness temperatures at 11 μm and 12 μm and their differences?

l128: ...)-2.1 μm...  : not clear

l133-135: it seems a bit redundant with l104-105.
l134: ...was intended... is it still a confidence level?
maybe add a reference for this SDS

l160: ...to that applied... replace by ...to the one applied...
        ...rather than consider**ing**....

l180: ...we use a naive definition of multilayer clouds here...
maybe say that, in a first step , we use a naive...  Otherwise I find it confusing as you
previously underlined the importance of this definition (l72-73)

l285: when you describe Fig8, say something about the liquid case.

l291:  around

l307: the sentence is not clear.

l315-316: the sentence should be rewritten

l354: if replace by it

**Figures**
General comments on the figures: please put the (a), (b)... labels out of the plots and
check the subtitles. Very often you repeat several times something that could be put in
the caption, and try to put explicit subtitles.

Also for the contingency tables, it would be useful to say somewhere that the numbers
are percentages of a population.

On several figures the labels for the x-axis are vertical, which is not convenient for
the reader, could you try to put them horizontally?

Fig1: MODIS MYDO6 C6.1 2008 : no need to write this 8 times
add some spaces between the plots, put bigger (a), (b)...

Fig2: caption: (b)  numbers ...
                 replace by identified
                ...**of** less than

Fig3: caption ...with (a) and without (b) the Pavolonis...

Fig4: P(MODIS...) is useless
MODIS COT >0.4 can be put in the caption.
caption : with (a) and without (b)

Fig8-9-10: I would do subplots:  (a) MODIS C6 liquid, (b) MODIS C6 ice.

---

## Referee Comment (RC3) · Anonymous Referee #4 · 25 Feb 2020

Review of 'Evaluation of the Aqua MODIS Collection 6.1 multilayer cloud detection algorithm through comparisons with CloudSat CPR and CALIPSO CALIOP products' by Marchant et al.

This manuscript by Marchant et al. shows an interesting evaluation of the MODIS/Aqua multi-layer product through comparisons to a lidar-radar cloud detection product from CALIPSO-CloudSat. These comparisons are thoroughly done for different multi-layer conditions, i.e. depending on the thermodynamic phase, optical depth and distance between the two layers. The performance of two versions of the MODIS algorithm (including the operational C6) are discussed, in terms of absolute accuracy against active products but also in terms of significance to avoid and/or flag biases in other MODIS cloud products.

The manuscript is clear, well written, of good scientific significance and absolutely fits the scope of AMT. I therefore advise for publication of this work, once the authors will have addressed the following (relatively minor) comments.
* * *
General comments:

1. Are the findings shown in this manuscript really limited to Aqua? I realize that the evaluation only is possible for the instrument onboard Aqua but is there a reason to think the conclusions are not just as valid for MODIS/Terra? If not, I wouldn't emphasize the Aqua dependence in the title and abstract.

2. The quality of the 2B-CLDCLASS-lidar product used in this study should at least be briefly discussed. For instance, I assume that the identification by lidar-radar of the thermodynamic phase becomes increasingly less accurate for the lowermost detected layers – is that of significance for the results presented here? Also, please explicit what is meant by "mixed phase".

3. It would be useful for readers and MODIS users if the authors further relate their results to the actual Cloud Multi Layer Flag SDS. In section 3 the authors describe the 4 methods / tests for multi-layer detection and explain that they are merged into a single confidence-level metric that ranges from 2 to 10 in case of multi-layer. I think that a couple more sentences explaining how the cumulative weight is obtained would be helpful. I realize that this paper does not aim to be too technical or replace the ATBD but it will likely become a reference paper for those interested in the multi-layer detection product. Also, it is unclear how the MODIS multi-layer cases that are shown in the manuscript actually relate to the SDS value, do they correspond to all cases with a value greater or equal to 2?

4. Related to the previous comment, and because this paper is likely to become a reference for the C6 multi-layer product, it would be very helpful if the authors included a brief bullet list of the practical implications of their findings, which users could easily refer to. For instance reminding that i) the MODIS multi-layer detection is to primarily be used as a retrieval quality indicator, ii) the flag should mainly be used when interested in ice cloud retrievals (as liquid cloud retrievals are by construction not too impacted?), iii) perhaps a word on the SDS values to be used (2 or higher?) for different cases, etc.
* * *
Specific comments:

1. p. 6 l. 145: Do I understand correctly that the L2 product in C6 includes the PH04 but the corresponding L3 product does not? If so, it would be worth emphasizing this by repeating it somewhere that be more visible to the readers (introduction or conclusion).

2. Fig. 3 and its analysis: It is interesting that the proportion of true/false detection of multi-layer cases in MODIS remains the same with or without using PH04. In both cases there is a 50% agreement with 2B-CLDCLASS and only the overall proportion of multi-layer detection changes. Would you then consider that PH04 does not significantly improve the quality of multi-layer detections or does the 8 vs 12% detection rate still make a difference to avoid biases on cloud properties? Fig. 11 indicates that PH04 does improve a bit the agreement ice cloud CER retrievals obtained in single- and multi-layer conditions, but I wonder if it is significant enough to risk higher false rejection rates.

3. Fig. 8 and its analysis: Why not also use the OD > 4 threshold here, for a better consistency with the following results related to Fig.. 9-11?

4. p. 13 l. 303–305: It is typically considered that effective radii retrievals associated with optical depth below 3 or 4 are not accurate, then is it really worth showing and discussing the results of Fig. 11?
* * *
Technical comments:

1. p3 l53: "a two-layer cloud overlapping model" sounds like the layers are not vertically separated, which would be surprising. Perhaps "a two-layer model" is sufficient?

2. p5 l105: might be worth precising "thermal IR".

---

## Short Comment (SC1) · 17 Mar 2020

Marchant et al. present a comprehensive evaluation on the performance of MODIS C6.1 multilayer cloud detection product using active observations. The evaluation is really helpful for better understanding and detecting multilayer clouds using passive instruments, and the manuscript is well organized and written. I have a minor suggestion that may be considered by the authors.

Multilayer cloud fraction may be one of the simplest and most important outputs for

the multilayer cloud product, while its accuracy is not directly compared with active observations. Our recent study (https://doi.org/10.1016/j.rse.2019.02.024) develops a multilayer cloud detection algorithm for VIIRS with the short infrared channels considered. Although the algorithm cannot be directly applied for MODIS due to the missing of the 2.25 micron channel, our discussion may be helpful to extend this work. In our study, we evaluated not only our results but also MODIS results. We found that the MODIS C6.1 can only detects slightly over 60% multilayer clouds (see Figure 9 of our work), and is this consistent with your conclusion? Thus, I suggest to present an evaluation of hit rate similar to Wang et al. (2019). Of course, this is optional considering that the results presented in this study have already been quite interesting and useful.

---

## Author Comment (AC1) · 17 Apr 2020

Anonymous Referee 2:

General Comments:

Thanks a lot for your comments and suggestions. Regarding the fraction of clouds where MODIS found multilayer clouds while active sensors not, I totally agreed with you that it depends on how multilayer clouds are defined and there could be some underlying sensitivity to the multilayer tests to cloud vertical structure within single con-

tiguous layers. This analysis is limited however to the definition of multilayer clouds used in MYD06 products (which assumes two separate cloud layers). But I agree that it could be very interesting to extend this analysis (in a future work since it is going to require a lot of data processing first) using 2B-CWC-RO and 2B-CWV-RVOD and a broader definition of multilayer clouds.

Note: To make this research reproductible, a Jupyter (python 3) Notebook has been created allowing to re-create all the figures of the paper and to download the data used: https://www.science-emergence.com/Jupyter/MODIS_myd06_collection_6_multilayer_clouds_analysis/View/

Answers:

⇢ The 'intent' of the algorithm is touched on at line 59 and lines 97-99. The latter occurrence seems out of place and would fit better near line 59. In fact, the 'intent' should be articulated for other multilayer algorithms besides MODIS.

Content has been updated accordingly.

⇢ Lines 77 and 79, references should have years added:

Years have been added.

⇢ Line 95, section should be two, not three

Section number have been updated

⇢ The discussion of figure 1 starting at line 138 is a little bit disjointed. I wasn't sure if panel (b) should be the sum of panels (e) to (h), or whether multiple positive tests can occur in a single pixel. I'm pretty sure it's the latter but it needs to be laid out clearer than is.

Yes, panel (b) is a combination of panels (e) to (h) (and each test does not have the same weight). So multiple positive tests can occur in a single pixel. Figure caption has been updated.

[Figure]

• To be clear, the Pavolonis and Heidinger algorithm is available within the L2 products but not in the L3 products? If that is correct, why is that the case?

Yes, it is correct the Pavolonis and Heidinger multilayer cloud detection algorithm output is available in L2 (through the MYD06 multilayer cloud QA) but it is not used for aggregating the MYD06 cloud products available in L3, since preliminary analysis during MYD06 Collection 6 development have shown that this algorithm was flagging too much cloudy pixels as multilayer clouds (this issue has been addressed in the MYD06 Collection 6 User guide).

• line 212, (a) and (b) should appear before including and excluding, respectively

Done

• line 252, answer about

Done

• figures 4 and 7 appear to have problematic axes. The number spacing in both axes is not uniform. Perhaps there is a rounding issue at play or the axes need to have additional bins or tick marks.

Thanks for noticing that, the issue comes from the grid which was 9 by 9, instead of 10 by 10.

• Figures 8 to 11, would be helpful to make clearer in each column at the top that this is "liquid" and "ice", or perhaps "liquid 2.1 um", "liquid 1.6 um", etc. The subpanel titles are pretty useless and could be included in the figure caption.

The subpanel titles have been removed and x-axis labels have been replaced by "liquid 2.1 um", "liquid 1.6 um", to make CER histograms easier to read.

• Furthermore, it would be easier to read the paper if cloud optical thickness reduced to the tau symbol or COT, and likewise with cloud effective radius could be r_e or CER

Cloud effective radius and cloud optical thickness have been replaced by CER and COT respectively in the paper main content.

---

## Author Comment (AC2) · 17 Apr 2020

Thanks a lot for your comments. We updated the figures following your suggestions and hope it looks better now.

Note: To make this research reproductible, a Jupyter (python 3) Notebook has been created allowing to re-create all the figures of the paper and to download the data used: https://www.science-emergence.com/Jupyter/MODIS_myd06_collection_6_multilayer_clouds_analysis/View/

[Figure]

- section III: It's not clear to me if there are some changes in the MODIS ML algorithm between C6 and C6.1 although it might be worth to explain somewhere briefly the differences between the 2 collections

MODIS MYD06 multilayer clouds algorithm is the same between C6 and C6.1. So C6.1has been replaced by C6 only (since the conclusions of the paper should be valid for C6 and C6.1 as well).

- l240: About the ML clouds ice/ice identified as liquid by MODIS, do you have an idea why?

We believed that it might be due to the ice cloud effective radius tests (used in the MODIS MYD06 C6 cloud thermodynamic phase algorithm) which have been trained using monolayer clouds only according to CALIOP 01 and 05 cloud layer products.

- l322: In the end, would you recommend to keep this PH04 test for the MODIS ML algorithm?

Yes, since the PH04 test contains useful information that can still be used for instance to filter MODIS MYD06 cloud effective radius (which is the primary goal of the MODIS MYD06 ML algorithm: to detect ML that can impact the cloud optical retrievals).

- Description of Fig2: is the product shown on Fig2b an official product? You do not mention or describe it in the paper. Is the 3km distance a common threshold to identify different layers?

Figure 2b does not show an official product, it is quick visualization that has been created to illustrate the impact of choosing an separation distance threshold to define multilayer clouds.

- l37: ...layers may strongly... replace by can, we are sure the presence of ML clouds can impact the retrievals

Done

- l49: I think the POLDER ML detection technique uses polarized reflectances but is not based on them.

Content has been updated

- l84-85: the sentence is not nice.

Content has been updated

- l107: and in the C6/C6.1

Done

- Globally: when you write 0.94 $\mu$m, like l20, there should be a space between the number and the unit, in Latex there is something similar to half a space (\, for me)

Done

- l123 to 125: not clear, do you mean : reflectances at 0.65 $\mu$m, 1.6 $\mu$m, 1.38 $\mu$m as well as brightness temperatures at 11 $\mu$m and 12 $\mu$m and their differences?

Yes, content has been updated

- l128: ...)-2.1 $\mu$m... : not clear

Sentence has been changed

- l133-135: it seems a bit redundant with l104-105. l134: ...was intended... is it still a confidence level? maybe add a reference for this SDS

Done

- l160: ...to that applied... replace by ...to the one applied... ...rather than considering....

Done

- l180: ...we use a naive definition of multilayer clouds here... maybe say that, in a first step , we use a naive... Otherwise I find it confusing as you previously underlined the

importance of this definition (l72-73)

Done

- l285: when you describe Fig8, say something about the liquid case.

Done

- l291: at effective radius around

Done

- l307: the sentence is not clear.

Sentence has been replaced

- l315-316: the sentence should be rewritten

Done

- l354: if replace by it

Done

- Figures General comments on the figures: please put the (a), (b)... labels out of the plots and check the subtitles. Very often you repeat several times something that could be put in the caption, and try to put explicit subtitles.

Done

- Also for the contingency tables, it would be useful to say somewhere that the numbers are percentages of a population.

Done, percentage % symbol has been added to each contingency table

- On several figures the labels for the x-axis are vertical, which is not convenient for the reader, could you try to put them horizontally?

The x-axis labels have been put horizontally now for figure 5 and 6.

- Fig1: MODIS MYDO6 C6.1 2008 : no need to write this 8 times add some spaces between the plots, put bigger (a), (b)...

Done

- Fig2: caption: (b) the numbers ... found replace by identified ...of less than

Done

- Fig3: caption ...with (a) and without (b) the Pavolonis...

Done

- Fig4: P(MODIS...) is useless MODIS COT >0.4 can be put in the caption. caption : with (a) and without (b)

Done

- Fig8-9-10: I would do subplots: (a) MODIS C6 liquid, (b) MODIS C6 ice.

Subplot titles have been removed and x-labels simplified to make the figure easier to read.

---

## Author Comment (AC3) · 18 Apr 2020

Thanks a lot for your comments and suggestions.

Updated figures can be found here: https://www.science-emergence.com/Jupyter/MODIS_myd06_collection_6_multilayer_clouds_analysis/View/

- 1. Are the findings shown in this manuscript really limited to Aqua? I realize that the evaluation only is possible for the instrument onboard Aqua but is there a reason to think the conclusions are not just as valid for MODIS/Terra? If not, I wouldn't emphasize

the Aqua dependence in the title and abstract.

You are right, the conclusions of the paper should also be valid for Terra as well. So, the Aqua dependence has been removed from the title and abstract. Same thing, I have changed C6.1 to C6 since the conclusions should also be valid for C6 and C6.1.

- 2. The quality of the 2B-CLDCLASS-lidar product used in this study should at least be briefly discussed. For instance, I assume that the identification by lidar-radar of the thermodynamic phase becomes increasingly less accurate for the lowermost detected layers – is that of significance for the results presented here? Also, please explicit what is meant by "mixed phase".

A couple of sentences have been added to the text to describe briefly the 2B-CLDCLASS-lidar product.

- 3. It would be useful for readers and MODIS users if the authors further relate their results to the actual Cloud Multi Layer Flag SDS. In section 3 the authors describe the 4 methods / tests for multi-layer detection and explain that they are merged into a single confidence-level metric that ranges from 2 to 10 in case of multi-layer. I think that a couple more sentences explaining how the cumulative weight is obtained would be helpful. I realize that this paper does not aim to be too technical or replace the ATBD but it will likely become a reference paper for those interested in the multi-layer detection product. Also, it is unclear how the MODIS multi-layer cases that are shown in the manuscript actually relate to the SDS value, do they correspond to all cases with a value greater or equal to 2?

In the manuscript MODIS multilayer cases relate to the MYD06 SDS value with a value greater or equal to 2 and the MYD06 1km Quality Assurance is also used to extract the PH test. Couple more sentences have been added to explain it.

- 4. Related to the previous comment, and because this paper is likely to become a reference for the C6 multi-layer product, it would be very helpful if the authors included

a brief bullet list of the practical implications of their findings, which users could easily refer to. For instance reminding that i) the MODIS multi-layer detection is to primarily be used as a retrieval quality indicator, ii) the flag should mainly be used when interested in ice cloud retrievals (as liquid cloud retrievals are by construction not too impacted?), iii) perhaps a word on the SDS values to be used (2 or higher?) for different cases, etc.

You are right, it is a good suggestion. The conclusion has been updated to better highlight the practical implications of this analysis.

- 1. p. 6 l. 145: Do I understand correctly that the L2 product in C6 includes the PH04 but the corresponding L3 product does not? If so, it would be worth emphasizing this by repeating it somewhere that be more visible to the readers (introduction or conclusion). 1 2.

Yes, it is correct the Pavolonis and Heidinger multilayer cloud detection algorithm output is available in L2 (through the MYD06 multilayer cloud QA) but it is not used for aggregating the MYD06 cloud products available in L3, since preliminary analysis during MYD06 Collection 6 development have shown that this algorithm was flagging too much cloudy pixels as multilayer clouds (this issue has been addressed in the MYD06 Collection 6 User guide).

- Fig. 3 and its analysis: It is interesting that the proportion of true/false detection of multi-layer cases in MODIS remains the same with or without using PH04. In both cases there is a 50% agreement with 2B-CLDCLASS and only the overall proportion of multi-layer detection changes. Would you then consider that PH04 does not significantly improve the quality of multi-layer detections or does the 8 vs 12% detection rate still make a difference to avoid biases on cloud properties? Fig. 11 indicates that PH04 does improve a bit the agreement ice cloud CER retrievals obtained in single- and multi-layer conditions, but I wonder if it is significant enough to risk higher false rejection rates.

I think it is a tricky question: It really depends on how multilayer cloud is defined and for

what purpose. The MODIS MYD06 multilayer cloud algorithm was first developed to detect only multilayer clouds (based on the assumption of two separated cloud layers) that will impact CER and COT retrievals (which are based on a homogenous monolayer cloud model) and not to detect all possible multilayer clouds from a passive sensor. The PH ML algorithm was designed to detect all multilayer cloud (regardless the impact of CER and COT). So, for MODIS MYD06 multilayer cloud the goal was first to determine if the assumption of a homogenous monolayer cloud model is good or not.

- 3. Fig. 8 and its analysis: Why not also use the OD > 4 threshold here, for a better consistency with the following results related to Fig.. 9-11?

I still think it could be interesting for a user to have at least one figure that provides an overview of the MODIS MYD06 CER distributions discriminated by CLDCLASS-Lidar monolayer and multilayer clouds.

- 4. p. 13 l. 303–305: It is typically considered that effective radii retrievals associated with optical depth below 3 or 4 are not accurate, then is it really worth showing and discussing the results of Fig. 11?

Yes there are large uncertainties on CER retrievals for OD lower than 3-4 but since there are some differences between CER distributions it can still be worth it to present them.

- 1. p3 l53: "a two-layer cloud overlapping model" sounds like the layers are not vertically separated, which would be surprising. Perhaps "a two-layer model" is sufficient?

Yes, you are right, a two-layer model should be sufficient. The content has been updated.

- 2. p5 l105: might be worth precising "thermal IR".

Thermal IR has been added to the text

---

## Author Comment (AC4) · 5 May 2020

Thank you Jianjie for your comment and suggestion on the paper. Evaluation of multilayer clouds are indeed a difficult problem since output statistics depend on how multilayer clouds are first defined. This is why we choose to present in the paper how multilayer cloud hit rate changed according to the upper layer cloud optical thickness and the separation distance between the cloud layers.
* * *